# Combined Antenna-Channel Modeling for the Harsh Horse Hoof Environment

**DOI:** 10.3390/s22186856

**Published:** 2022-09-10

**Authors:** Jasper Goethals, Denys Nikolayev, Arno Thielens, Günter Vermeeren, Leen Verloock, Margot Deruyck, Luc Martens, Wout Joseph

**Affiliations:** 1IMEC-WAVES, Ghent University, 9000 Ghent, Belgium; 2IETR (l’Institut d’Électronique et des Technologies du Numérique), UMR 6164, CNRS, Université de Rennes, 35000 Rennes, France

**Keywords:** electromagnetic propagation, microstrip antenna, animal telemetry, Internet of Animal Health

## Abstract

This paper describes the exploration of the combined antenna–channel model for a horse hoof. An antenna of 25 mm × 40 mm is designed in the ISM 868 MHz band. During the characterization and design of the antenna, the dynamic and harsh environment of the horse hoof is taken into account throughout every step of the procedure because it is impossible to de-embed the antenna from its environment. The antenna and channel model are verified extensively by measurements in phantom and ex vivo. The antenna is verified to be robust against changes in the morphology of the horse’s hoof up to 50%. The dynamic environment was captured by considering different soil types and air, and the design was verified to be resilient against changes herein. The antenna performs well within the targeted band, with a fractional bandwidth of 8% and a gain of −2 dBi. Furthermore, a path loss model was constructed for a typical barn environment, and the antenna reaches a range of 250 m in the studied environment based on the LoRa technology. This research is important for monitoring horse health.

## 1. Introduction

Internet of Things (IoT) enabled an invaluable new set of tools in many fields [1]. For farmers and veternarians, the applications of IoT were most prominent in the Internet of Animal Health (IoAH) [2]. The earlier an anomaly in health and fitness is detected, the higher the chances for a full recovery [3]. To monitor the horse’s vital parameters, one can put a monitoring device or a wearable on the animal, which will inform the caretaker at an early stage about any possible anomaly in these parameters [4]. This requires 24/7 logging of data [5,6], which can be transmitted using a wireless connection from the device to a gateway, e.g., installed in the stable. There are a series of interesting vital parameters such as movement, heart rate, temperature, and lactates, that can be monitored using wearables or implants. Today, equine wearables mainly focus on detecting movement (for training purposes) and behavior (for health purposes) by using an accelerometer attached to the horse’s head or torso [7,8]. Unfortunately, a horse can never be equipped 24/7 with such a device, since a wearable attached to head or torso can not only harm the horse (chafing, tendon damage, etc.) or hinder movement (especially during training), but is also prone to damage by the horse (e.g., biting, rubbing against the wall). One way to address this issue is to place a wearable under the hoof inside a thin pad like a sole in a shoe (made of leather or silicone). This pad is attached to the hoof through the nails of the horseshoe and feels natural for the horse since these pads are today often used to solve orthopedic issues and remain on the horse for a period of 6 weeks without disturbing its behavior. The choice for an in-hoof device does have a disadvantage since it limits the number of bio-signals that can be obtained to accelerometer data. Core-temperature and heart rate can only be measured by a wearable on the horse’s body, which implies that these devices are faced with the same problems as mentioned above (harming and/or hindering horse, damage by the horse). Lactates can only be measured in the body through blood values and is out of the scope of any wearable solution.

Figure 1a shows a horseshoe attached with a hoof pad. Note that the frog is considered to be the hoof’s weak part and cannot be covered by any electronics, hence limiting the size of the electronics and antenna. The goal of this study is to design a wireless off-body telemetry device that operates from inside the hoof and sends the data to a cellular gateway. To cover sufficiently long ranges during training and recreational activities and in order to communicate directly from the horse to a gateway, radio technologies with suitable propagation characteristics need to be used. Lower frequency waves are typically associated with larger achievable ranges. However, lower radio frequencies typically require larger antennas and limit the maximum achievable throughput due to smaller available bandwidths, thereby limiting device placement options and the amount of monitoring data, respectively. Here, we choose the Industrial, Scientific and Medical/Short Range Devices (ISM/SRD) 868 MHz band. This opens the possibility to use radio technologies such as Longe Range Wireless Antenna Network (LoRaWAN), Sixfox, and other low-power IoT technologies. To this aim, the channel of the configuration is characterized, and a link budget is calculated based on simulations. Conductive objects (such as horseshoe) located in the near field strongly affect the antenna radiation performance [9,10]. In addition, proximity of dielectric objects (horse’s hoof) impact both the impedance and radiation performances [11,12]. The physical channel for off-body communication (from the sensor in the hoof to the private/cellular operated gateway) is required in order to assess the range of the wearable. Most research on off-body propagation is focused on humans [13,14]. However, other studies concluded that radio performance will be significantly affected by an animal’s body [15,16]. Kwong et al. [17] proposed a simple two-path model for off-body communication based on data obtained by numerical calculations. Benaissa et al. [18,19] investigated on-to-off-body path loss and communication for dairy cows in stables at 2.4 GHz and 868 MHz. In this work, we explore an antenna design approach constrained by the aforementioned factors of operation in the harsh and dynamic environment of a horse hoof at 868 MHz. This paper addresses the design challenges of the antenna in the horse pad, experimental validation using a phantom and ex vivo, and channel characterizaton. This work builds upon previous work [20,21]. There are some crucial differences, this work has (i) an adapted antenna design, (ii) a better understanding about the horse hoof environment, and (iii) presents a path loss measurement campaign. During the measurements, the robustness to the changing environment of the antenna was both simulated and measured. The novelty of this work is the ex vivo measurements for both the antenna performance and the channel characterization, i.e., (i) the experimental design and verification of the simulated results on a real horse leg and (ii) characterization of the wireless channel and off-horse link budget for network planning for real scenarios. Additionally, validation measurements are performed within the realistic setting of a barn rather than only considering simulated environments.

This paper is organized as follows: Section 2 focuses on the antenna design considerations and the scenarios incorporating the moving aspects, the modeling approaches, and the materials. Section 3 reports the phantom and ex vivo measurements and validation. The link budget calculations are presented in Section 4. Section 5 concludes the paper.

## 2. Planar Inverted-F Antenna Design and Phantom Model

### 2.1. The Antenna Design

The design has to comply with the spatial constrains of the device, needs to be safe for the horse, and needs to use materials that are biocompatible. In this work, the antenna is embedded in a large hoof pad with dimensions 135 × 150 × 4 mm${}^{3}$. A hoof pad is usually a thin sheet of leather, plastic, or silicon used to prevent rocks or uneven a hard surface from hurting the bottom of the hoof. A border of around 2 cm has to be kept clear around the edge of the hoof, since there the horseshoe will press the pad against the hoof. Between the pad and the hoof, an elastic epoxy is applied. This fills the cavity constructed by the pad and the hoof. The epoxy is applied to prevent dirt and rocks from agitating the sensitive hoof area while wearing the pad and horseshoe. The thickness of this layer depends on the horse type. For example, a flat-footed horse will not form this cavity. The antenna needs to be embedded into these layers in order to prevent contact between the horse’s body and the electronics and ensure biocompatibility of the antenna. A schamatic abstraction of the hoof setup is shown in Figure 2.

The parameters of interest are the reflection coefficient, −10 dB bandwidth (BW), the center frequency, the radiation efficiency, and the gain of the antenna. Based on the geometrical limitations of the space in the hoof pad (height < 0.01 λ), antenna types that are suitable for the considered applications should be of a low profile. Additionally, the avoidance of extremities in the antenna’s shape, for example a conducting wire orthogonal to the horse’s sole that might protrude in the body under pressure, is another argument for working with a planar structure. Within the planar design options, we considered the printed inverted-F antenna design (IFA) and a printed loop antenna. The loop antenna was hard to match to the reference impedance, and more energy would be dissipated in the matching network. We did not consider a patch design (i.e., two-layered design with ground layer) because this would be too directional for our use case. When the horse would lift its leg and turn it 90 degrees, as it does during a movement, a patch antenna would radiate along the surface of the soil towards one side only. This was unwanted behavior. Due to the given area on the PCB, a miniaturization technique had to be employed. The most efficient according to our exploratory study (not included in this work) was a meandered strip-line solution. Capacitive loading was also considered but deemed too inefficient in terms of radiation. Furthermore, the printed IFA can be easily manufactured in a mechanically robust configuration since it does not require vertical shorting strips or vias that reduce the rigidity of the structure. Figure 1b shows the antenna layout based on our previous paper [20,21]. The feed placement can be used to match the antenna to 50 Ω of the RF frontend. The dimensions of the board are 40 × 25 × 1.55 mm${}^{3}$(corresponding to 0.12 × 0.07 × 0.01 $\lambda^{3}$), hence fulfilling the constrains of the device. The antenna itself is 12 × 25 mm${}^{2}$. The substrate material for a prototype is FR-4 ($\varepsilon_{r}$ = 4, σ = 0.0038 S/m) [22]. However, more durable and shock resistant substrates with comparable electromagnetic (EM) properties could be employed. All metallization is modeled as a perfect electrical conductor (PEC). The dimensions are denoted on the schematic of the antenna in Figure 1c. During the design, the environmental properties and elements were incorporated and studied one by one to quantify their effects on the antenna performance.

### 2.2. Application Scenarios

The environment of a horse’s leg is dynamic. When the horse is in a meadow, it will not only graze, but also walk, trot, gallop, and it might even buck. Since the monitoring device is placed in the hoof pad, the antenna environment will also change dynamically. In this paper, two main scenarios are considered; Figure 3a illustrates both scenarios. In the first case, the horse has his hoof lifted off the ground. The angle of the front hoof in the air is determined to be 90 degrees [23] with respect to the ground and lifted 10 cm during a normal walk (so called working step). During this first raised scenario, the leg will be in free space; thus the effect of the soil on the antenna near field could be neglected. In the second scenario, the hoof is on the ground. The soil is modeled by a 500 × 500 × 50 mm${}^{3}$ block. In this study, we consider grass and concrete as a soil. Grass was assigned a relative permittivity $\varepsilon_{r}$ = 8 and a conductivity σ = 0.163 S/m [24], whereas concrete was assigned $\varepsilon_{r}$ = 4.5 and σ = 6.25 mS/m [25]. Furthermore, the epoxy layer thickness will vary from horse to horse. To study the dependence of this layer, different thicknesses were investigated.

### 2.3. Numerical Phantom Model

The antenna is embedded in a pad that consists of polylactic acid (PLA) and epoxy ($\varepsilon_{r}$ = 3.5, σ = 0.0023 S/m) [26]. In the simulations, the PLA and the epoxy have the same EM properties. This encapsulation increases the robustness of the parameters of the antenna with reference to the influence of the horse leg and the ground and is necessary for the mechanical stability of the pad. Figure 2 shows the complete configuration. A horse’s leg is simulated by a 10-cm-long cylinder with diameter 15 cm. The complete setup is built of the four layers: the horseshoe, the pad, the filling epoxy, and the leg. We assume that the leg is made out of bone (homogeneous) (($\varepsilon_{r}$= 12, σ = 0.18 S/m) with parameters similar to human bone [15]. The horse’s hoof consists mostly of bone; therefore, this tissue has been selected. However, another simulation with the horse leg out of muscle ($\varepsilon_{r}$= 51, σ = 0.98 S/m) [15] is performed to assess the effect of the variation of environment EM properties. The horseshoe is modelled as PEC. The antenna is tuned to the scenario where the leg is in the air and is made out of bone, and the filling epoxy is 5 mm thick. The numerical simulation is performed using Sim4Life (S4L), a finite-difference time-domain (FDTD) simulator [27]. For the calculation of the BW and center frequency, a Gaussian signal was excited over the edge source at the feed strip. The center frequency of this Gaussian signal is 1 GHz, and the BW is 1 GHz. The soil block terminates into perfectly matching layers. In the second step, the simulations are performed with a harmonic excitation at 868 MHz to calculate the efficiency, radiation characteristics, and the reflection coefficient at the center frequency of operation. The number of cells of the grid were 2.3 MCells and 2.8 MCells for the BW simulations without or with ground, respectively, and 1.8 MCells and 2.0 MCels for the gain pattern simulations without or with ground, respectively.

### 2.4. Numerical Results

The first configuration is for the lifted hoof, and Figure 4a shows the reflection coefficient of the antenna. Table 1 summarizes the results. The simulation shows that the bone setup is well matched for the aimed band. Different leg tissues and epoxy thicknesses will shift the resonance due to change of the effective dielectric permittivity. Table 1 shows that the efficiency lowers significantly with a muscle phantom, since this material has a higher conductivity in comparison to bone. The lower efficiency of <30% is due to the dense tissue environment around the antenna. Furthermore, the thickness of the filling epoxy has a significant influence on the tuning of the antenna. Figure 4a shows that the antenna is robust against contained variations in the filling epoxy thickness with respect to the targeted band, namely the aimed band is still served with deviation of 2 mm in thickness. The influence of the epoxy variations are due to the significant difference in electrical parameters between bone and epoxy. The epoxy has a relative permittivity of 3.5 and a conductivity of 0.0023 S/m, this in contrast with the bone tissue parameters, which are 12 and 0.18 S/m, respectively. The more epoxy is present in the near-field of the antenna, the lower the effective permittivity will be and will thus result in a higher resonance frequency. Because of the lower conductivity of the epoxy, a thicker epoxy layer will result in a higher radiation efficiency. Figure 4b shows the influence of the different soil types. We can conclude that the antenna is robust against the change in soil type according to the simulations. Variations less than 0.01% and only a slight detuning (<0.1%) are obtained. This is necessary to achieve a stable horse hoof to gateway communication. This robustness to the soil types is explained by the air gap provided by the iron horseshoe. Due to the type of the antenna, a planar IFA, and its typical radiation pattern [28], the iron horseshoe has a negligible influence on the antenna performance.

## 3. Experimental Validation

### 3.1. Phantom Measurements

As an intermediate validation step, a phantom was built to validate the simulations. The phantom is depicted on Figure 5. It consists of the pad, a real iron horseshoe, and a polyvinyl chloride (PVC) tube as container for the phantom liquid, which was tap water. A small disk of epoxy resin of 10 mm height was used to replicate the filling epoxy; tap water was used to recreate the horse leg tissue. The main purpose of these measurements on a phantom were to show correspondence between measurements and simulations, not to emulate a real horse leg (see Section 3.2). The dielectric properties of the water used to fill the phantom are not the same as those of muscle and bone tissue that can be found in a real horse leg. The pad is constructed by 3D-printing a 15 cm diameter cylinder of 1 mm height and a border of 4 mm. A small holder for the 4 × 2.5 cm${}^{2}$ PCB is also printed with PLA. This pad is then filled with epoxy resin to finish the pad as a solid cylinder with the antenna PCB embedded. The produced pad is seen in Figure 5. A miniature X.FL [29] connection is included to connect the antenna to the test bench with a 50 Ω feedline. The reflection coefficient is measured with the vector network analyzer (VNA) Rohde and Schwarz ZNB 20.

### 3.2. Ex Vivo Measurements

To validate the simulation, measurements are performed on a real horse leg ex vivo. A leg of a recently deceased horse (<24 h) was severed and provided by the faculty of veterinary medicine at Ghent University. The PLA-epoxy hoof pad with the embedded antenna was attached using tape. Next, the hoof iron was attached over this ensemble with more tape. Between the hoof pad and the hoof, a cavity is formed due to the morphology of a horse hoof. This cavity was filled with epoxy Equi-pak [30]. The farrier usually does this to prevent dirt from entering this cavity and consequently prevent inflammation of the hoof tissue. In the test setup, the horse hoof was at a height of 64 cm from the ground and held by a wooden trestle, as depicted in Figure 3b. The measurements were performed at a surgical stable at the faculty of veterinary medicine. In addition to the free space measurements, different soil types were examined by pressing the hoof gently on the surface. Again, grass and concrete were considered. Figure 3b shows the different setups.

### 3.3. Experimental Validation Results

Figure 4c compares the results of the measurements on the real horse leg and the simulation of the case of the horse leg on the concrete floor. A good agreement between the simulations and the experiment is obtained (Figure 4c; <1% difference in resonance frequency). The measured bandwidth of the antenna is 825–898 MHz, which serves the ISM-868MHz band perfectly as depicted in Figure 4c. The mismatch is explained by the thickness of the filling epoxy and the unknown parameters of the horse leg. The offset in reflection coefficient between simulation and measurement is due to the loss in the X.FL wire. The robustness to the different soil types is confirmed and is shown in Figure 4d. The frequency shift is less than 3%, and the targeted band is well under −10 dB for all types of soil.

## 4. System Loss Measurements

### 4.1. Path Loss Model

System loss (SL) is defined as path loss (*PL*) added to cable loss and subtracted from antenna gain. Because the gain and cable losses are constant in function of distance, we will focus on the path loss. To model the off-body path loss, a single slope model is chosen [31]:(1)$$PL=PL_{0}+10n\log\left(\frac{r}{r_{0}}\right)+\chi$$

With *PL*${}_{0}$ the reference path loss at distance *r*${}_{0}$, *n* is the propagation coefficient, *r*${}_{0}$ the reference distance, and χ the zero-mean Gaussian distributed error with a standard deviation σ. This is a measure for the fading margin [32]. The reference point *r*${}_{0}$ is 3.5 m, which translates into ≈10λ. The data is binned with a bin size of 3 m to ensure independent samples for the statistical analysis. The model was fitted using the least square method. The received power is measured with the SRM 3006 spectrum analyzer with a complementary tri axial antenna by Narda Safety Test Solutions [33]. Each distance point is the average of 400 traces with 60 ms sweep time and a resolution BW of 1 kHz. The setup is shown in Figure 3c.

### 4.2. Radio Technology and Link Budget Calculations

The link budget was calculated using the LoRa technology specifications [34]. The sensitivity of the chip is inversely correlated with the data throughput. Some LoRa link budget values are listed in Table 2.

The spreading factor (SF) corresponds with the length in time of a symbol. The sensitivity (*S*) (in dBm) is than calculated as
(2)$$S\left[\mathrm{dBm}\right]=-174\mathrm{dBm}+10\log\left(\frac{\mathrm{BW}}{1\mathrm{kHz}}\right)+\mathrm{NF}\left[\mathrm{dB}\right]+\mathrm{SNR}\left[\mathrm{dB}\right]$$
with BW the bandwidth in kHz, NF the noise factor which is dependent on the hardware in decibels, and SNR the signal to noise ratio in decibels. The link budget also depends on the power of the transmitter. The maximum allowed power is 25 mW or 14 dBm in the 868 MHz ISM/SRD band [34]. The link budget will be calculated for the two cases: the horse leg lifted in the air and standing on the ground. The sensitivity will also depend on the data rate. SF corresponding sensitivity values are given in Table 2. The aimed application is a health monitoring system for horses. This on-edge classification system sends the status of the horse to a central server via a gateway located on the grazing field a few times a day. Due to the fact the device processes the movement on the fly, only the status has to be sent, and the required data transfer over the air is very limited. A data rate of 500 bps will be enough to service this application. Our measurement scenario presents a realistic, yet simple use case. In reality, use cases will be dynamic and will involve multiple users of the system, which can lead to interference and (multipath) fading. We assume that these effects can be mitigated at the access point, which is common in LORA systems [35,36].

### 4.3. System Loss and Link Budget Results

For the two cases, the measured system loss is fitted to the one-slope model, see Equation (Equation 1). The resulting propagation coefficients and fading margin are given in Table 3. The fitted model is shown in Figure 6. With LoRa as the considered RF technology, the ranges depending on the case and data rate are listed in Table 2. Here it is shown that the setup is able to reach the targeted range of 250 m with a data rate of 500 bps despite the harsh environment of the antenna. These are obtained using the method described in Section 4.1 and Section 4.2; namely, the system loss models of Equation (Equation 1) and Table 3 are used to determine the distance at which the received power becomes lower than the sensitivity in 95% of the cases, i.e., the range. The results are in line with previous literature [18,31]. The significant difference in system loss between the two cases can be explained by the fact that the two cases make use of a different sector of the radiation pattern with different directivity values.

## 5. Conclusions

In this paper, the combined antenna–channel model of a horse hoof is investigated using FDTD simulations, phantom measurements, and real horse leg experiments. The antenna cannot be de-embedded from its environment. Therefore, the channel was taken into account in every step of the design process. We present a planar IFA antenna with center frequency at 868 MHz which can be integrated into a hoof pad and is robust to changes in its channel such as different soils types and robust against different horse hoof morphologies. The path is modelled with a one slope path loss model for two different configurations of a horse’s leg. Based on this model and the specification of the LoRa technology, the range of the antenna is determined to be 250 m. This range suffices the application demand of a remote health sensor for grazing horses. Future work will consist of investigating the proposed design on a real living horse, studying the variation of different types of horses, investigating the use of supporting structures around the antenna to increase durability, and developing a real application of behavioral data to monitor the horse. We envision that this system will be integrated in fifth generation networks [37]. This system can also be used on other animals to measure other vital parameters, potentially while connected to other wearable and implantable devices and sensors installed in the stable or around the animal [38]. IoAH and IoT in general have mainly been serviced by the following technologies: cellular networks, WiFi, Bluetooth, IEEE802.15.4, LoRA and NB-IoT. The two largest challenges will be the massive amount of users/nodes, low power, and the security [38,39], for which the current technologies do not have the answer to yet. The next generation of communication will have to supply the resources to overcome these challenges [40]. Agricultural IoT will also bring challenges to other fields, such as big data, law, and biomedicine [41]. 

## Figures and Tables

**Figure 1 sensors-22-06856-f001:**
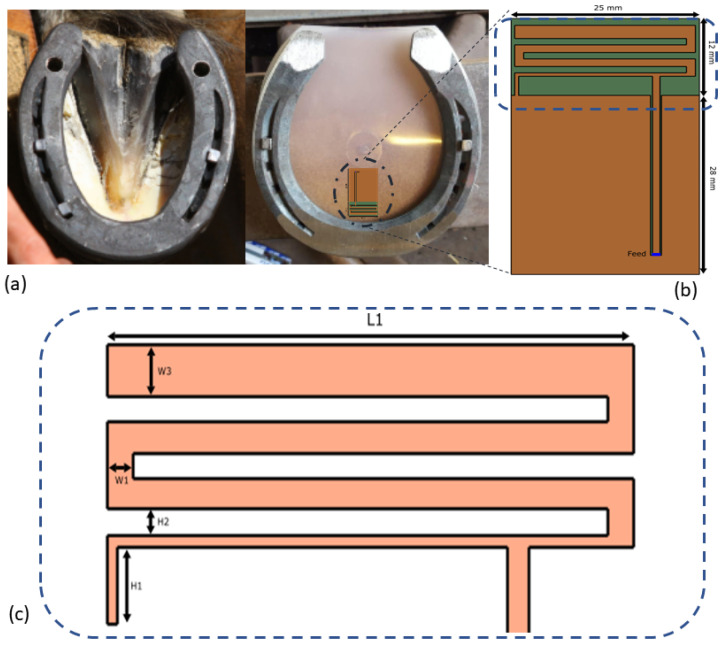
(**a**) An example of a horse hoof and a horse hoof with the planar pad with the embedded electronics: (**b**,**c**) design of the planar IFA antenna; the dimensions in (**b**,**c**) design of the planar IFA antenna. The dimensions in (**b**) are WT = 25 mm, LT1 = 12 mm, and LT2 = 28 mm; in (**c**) they are H1 = 3 mm, H2 = 1 mm, W1= 1.2 mm, W3 = 2 mm, and L1 = 25 mm, and the width of the short strip is 0.5 mm. The meandering arm continues with this thickness until the first corner after the feedline, thereafter it changes to W1 until the last corner to W3. The gaps between the meandering lines are all H2. The distance between the shorting pin and feed is 18 mm.

**Figure 2 sensors-22-06856-f002:**
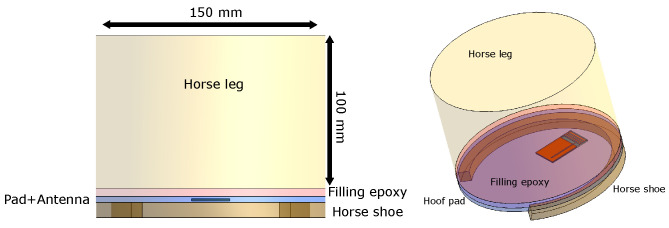
The model setup consisting of the horse leg, the hoof pad with the embedded antenna, the filling epoxy, and the horseshoe. Horseshoe = 10 mm, Pad+antenna = 4 mm, Filling epoxy = 5 mm.

**Figure 3 sensors-22-06856-f003:**
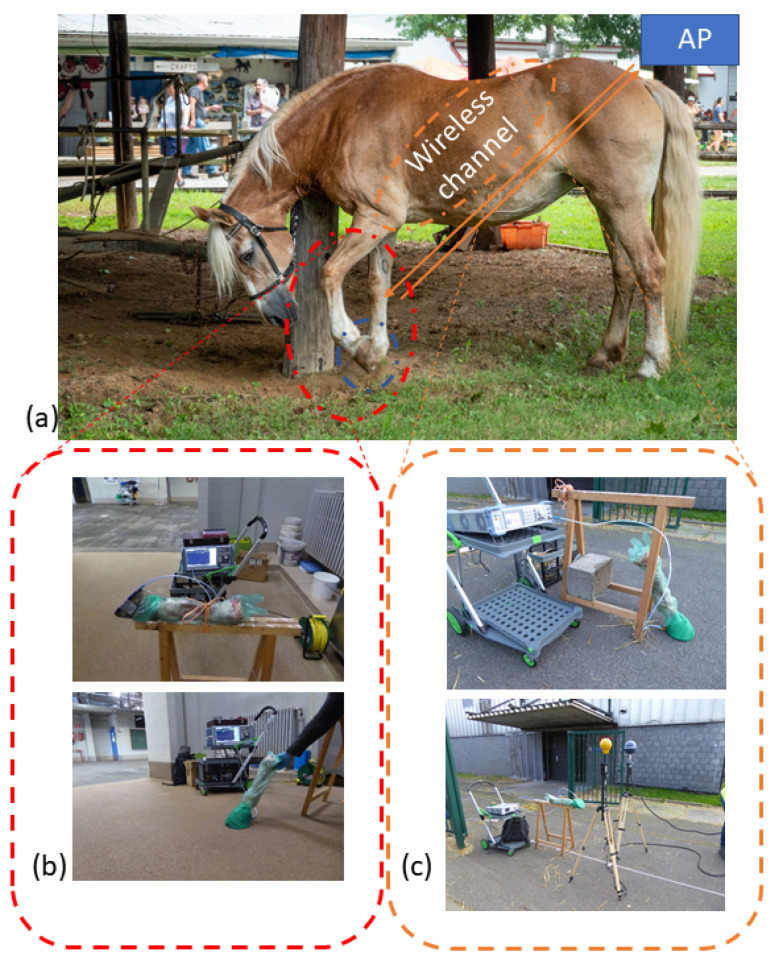
(**a**) the overall concept of the project; a mobile sensor embedded in the horseshoe communicating the behavioral data of the horse to a access point located somewhere in the pasture the horse resides. This work studies the combined antenna–channel modeling of this situation; (**b**) shows the constructed pad, the severed horse leg, covered in a hygienic veterinary glove; part of the hoof is uncovered by the glove and the hoof pad is attached to it with tape. The top picture is the air-case setup. The iron is attached by means of the green tape as seen on the ground-case setup, the bottom picture; (**c**) the path loss measurement setups as discussed in Section 4.1.

**Figure 4 sensors-22-06856-f004:**
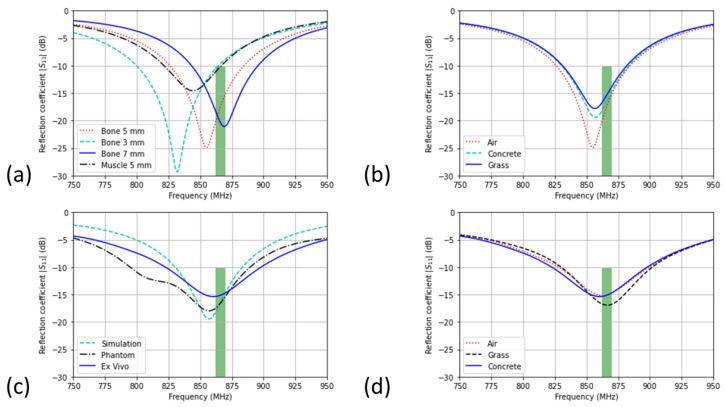
(**a**) Simulated reflection coefficient |S${}_{11}|$ for the antenna in air. To characterize the impedance robustness, the phantom is modelled as muscle or bone, and the epoxy layer is varied from 3 mm to 7 mm; (**b**) robustness of the hoof-antenna against the different soil types (simulated); (**c**) measured reflection coefficient using a phantom leg and an ex vivo leg; (**d**) measured reflection coefficient with the leg standing on different soil types.

**Figure 5 sensors-22-06856-f005:**
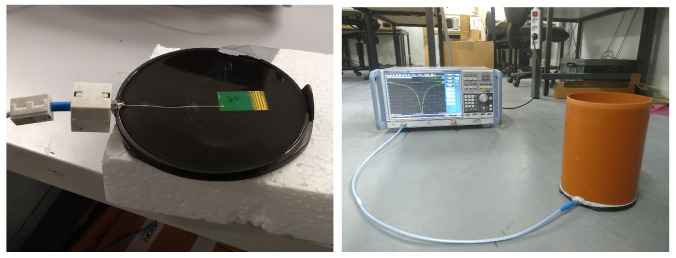
The embedded antenna in the epoxy disk representing the hoof pad connected to the VNA with an SMA-X.FL adaptor and the full setup on the ground.

**Figure 6 sensors-22-06856-f006:**
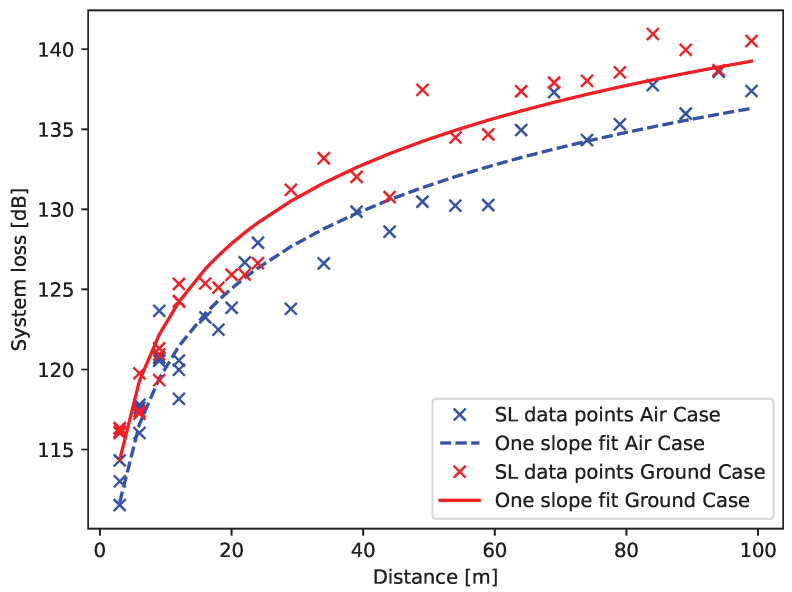
The fitted model (solid line) on the measured data (markers) for both propagation cases: in the air (blue) and on the ground (red).

**Table 1 sensors-22-06856-t001:** Simulated efficiencies in different scenarios.

Case	Tissue	Epoxy	Radiation	Max. Gain
		Thickness	Efficiency	$G_{0}$
Air	Bone	5 mm	21%	−2.4 dBi
Air	Muscle	5 mm	1.4%	−12.6 dBi
Ground (concrete)	Bone	5 mm	14%	−2.41 dBi

**Table 2 sensors-22-06856-t002:** Predicted ranges based on the developed path loss model and simulated antenna gain.

SF	BW	Bit Rate	*S*	R (Air)	R (Ground)
	[KHz]	[kbps]	[dBm]	[m]	[m]
7	500	21	−117	26	18
9	500	7	−123	63	43
11	500	2	−128.5	137	94
12	250	0.5	−134	301	204

**Table 3 sensors-22-06856-t003:** The propagation coefficient and fading margin for the air and ground case.

	Ref. System Loss	Path Loss Exp.	Fitting Error
	SL${}_{0}$ [dB]	n [-]	σ [dB]
Air	92.70	1.62	1.93
Ground	95.71	1.63	1.79

## Data Availability

Not applicable.

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
