# Peer review of "Combined Antenna-Channel Modeling for the Harsh Horse Hoof Environment"

_sensors, 2022, doi:10.3390/s22186856_

Round 1

Reviewer 1 Report

This paper is a combination of two previously published conference papers by the same authors, with some more details:
1. Design and Verification of an On-Body Antenna in the Harsh Environment of a Horse Hoof.
2. Combined Antenna-Channel Characterization 348 for Wireless Communication from Horse Hoof to Base Station.

This is a good piece of work and I recommend it for publishing in your journal.

At the same time, I have one concern; in the simulation, why the horse leg model does not consider both bone and muscles together, instead of two separate simulation, one with bone tissue and another with muscles tissue? 

Author Response

Thank you for your kind words. This is indeed a continuation and large extension of the two mentioned papers. The latest of which is also cited in the introduction (reference 18). I will add our first work for completeness.

To answer the question, the real constitution of a horse leg is mostly bone with some tendons [1]. Therefore we abstracted the horse leg to a bone cylinder without much loss of generality. Yet, we considered a leg consisting fully of muscle to examine the robustness of our antenna in an extreme edge case.

[1] B. Faramarzi, et al., “Morphovolumetric Analysis of the Hoof in Standardbred Horses,” Journal of Equine Veterinary Science, Volume 71, 2018, Pages 40-45.

Reviewer 2 Report

1) In the Introduction, a more deep review of how emerging IoT technologies are revolutionizing the farming (and more generally, the environment) field should be provided. Nowadays indeed, we are experiencing a rapidly increasing use of IoT technologies that, as the authors correctly highlighted, bring new invaluable tools. In this respect, some recent, more general and transversal references can be added and discussed in the Introduction to support the importance of such topics, such as:

- "Toward Integrated Large-Scale Environmental Monitoring Using WSN/UAV/Crowdsensing: A Review of Applications, Signal Processing, and Future Perspectives", Sensors, 2022;

- "A survey on the role of IoT in agriculture for the implementation of smart livestock environment", IEEE Access, 2022.

2) In the Introduction, it should be clearly highlighted (possibly with a bullet list) how the present contribution extends the previous work in [18] from the same authors.

3) The one-slope path loss model considered in eq. (1) is well known to be potentially inaccurate in environments characterized by severe negative propagation effects (e.g., dense multipath or shadowing). The experiments seem to be conducted in a rather controlled environment. Is it reasonable to assume the same link budget and accurate fitting results in a real farming environment, where harsh propagation effects such as multipath can be actually present? If so, how to cope with them?

4) Do the authors see any possibility to implement a similar system using one of the emerging 5G or beyond communication technologies? Are there any existing works on the topic? A more deep discussion on the future perspectives envisioned for the field would be useful.

Author Response

Thank you for taking the time to thoroughly reviewing our paper. I will answer your questions one by one.

  1. We would like to thank the reviewer for his literature suggestions and we added them on the relevant places in the text.

Internet of Things (IoT) brought for many field an invaluable new set of tools [1].  For farmers and veterinarians, the applications of IoT was most prominent in the Internet of Animal Health (IoAH) [2].”

With [1] the sensors paper and [2] the IEEE access paper.

  1. We made the difference more clearly by adding the following statement to the text. Due to personal preferences, with opted with in-text clarification in contrast to bullet-points. This summarizes what is lacking in the conference paper.

This work build upon on previous work [17], [18]. There are some crucial differences, this work has i) an adapted antenna design, ii) a better understanding about the horse hoof environment and iii) presents a path loss measurement campaign.

  1. The reviewer remarks that the path loss measurement was done in a controlled environment, which would lead to inaccurately extrapolating to a real life scenario. Yet we would like to counter this argument by noting that the experiments were conducted at a real horse stable using a real horse leg. For this reason, we suggest that any multipath and slow fading effect that would be encountered in a real scenario is accounted for in our model. This approach is also used in citation [17] of the manuscript.

To remediate the effects of multipath, slow fading, or multiple users interference, we would implement the access point in such way that these problems are solved. In our opinion, these problems should not be solved on the level of the low power wearable. This would significantly decrease its lifetime. There is sufficient literature on beam steering and other multiple-input multiple-output systems and high traffic solutions at access point level that we can apply to this use case.

To address this question, we added the following paragraph in the methodology of our path loss measurement:

“Our measurement scenario presents a realistic, yet simple use case. In reality, use cases will be dynamic and will involve multiple users of the system, which can lead to interference and (multipath) fading. We assume that these effects can be mitigated at the access point, which is common in LORA systems [36], [37].”

[17] Benaissa, S., Plets, D., Tanghe, E., Vermeeren, G., Martens, L., Sonck, B., Tuyttens, F., Vandaele, L., & Joseph, W. (2017). Internet of animals: On-and off-body propagation analysis for energy efficient WBAN design for dairy cows. 2017 11th European Conference on Antennas and Propagation, EUCAP 2017, 298–302. https://doi.org/10.23919/EuCAP.2017.7928112

[36] O. Georgiou and U. Raza, "Low Power Wide Area Network Analysis: Can LoRa Scale?," in IEEE Wireless Communications Letters, vol. 6, no. 2, pp. 162-165, April 2017.

[37] Jinping Niu, et al., “Energy-efficient adaptive virtual-MIMO transmissions for LoRa uplink systems,” Digital Signal Processing, Volume 127, 2022, 103493.

  1. To answer the question about the outlook on the next generation of wireless communication and its relevance to this work, we will add the following paragraph to the conclusion to indicate our future prospect about the agricultural IoT field.

“In the future, we envision that this system will be integrated in fifth generation networks [38]. This system can also be used on other animals to measure other vital parameters, potentially while connected to other wearable and implantable devices and sensors installed in the stable or around the animal [39].  IoAH and IoT in general has mainly been serviced by the following technologies: cellular networks, WiFi, Bluetooth, IEEE802.15.4, LoRA and NB-IoT. The two largest challenges will be the massive amount of users/nodes, low power, and the security [39][40], for which the current technologies do not have the answer to yet. The next generation of communication will have to supply the resources to overcome these challenges [41]. Agricultural IoT will also bring challenges to other fields, like big data, law and biomedical [42].”

With the following papers added to the bibliography:

[38] L. Chettri and R. Bera, "A Comprehensive Survey on Internet of Things (IoT) Toward 5G Wireless Systems," in IEEE Internet of Things Journal, vol. 7, no. 1, pp. 16-32, Jan. 2020, doi: 10.1109/JIOT.2019.2948888.

[39] K. Grgić, D. Žagar, J. Balen and J. Vlaović, "Internet of Things in Smart Agriculture — Possibilities and Challenges," 2020 International Conference on Smart Systems and Technologies (SST), 2020, pp. 239-244, doi: 10.1109/SST49455.2020.9264043.

[40] K. Grgić, A. Pejković, M. Zrnić and J. Spišić, "An Overview of Security Aspects of IoT Communication Technologies for Smart Agriculture," 2021 16th International Conference on Telecommunications (ConTEL), 2021, pp. 146-151, doi: 10.23919/ConTEL52528.2021.9495985.

[41] D. Wang, D. Chen, B. Song, N. Guizani, X. Yu and X. Du, "From IoT to 5G I-IoT: The Next Generation IoT-Based Intelligent Algorithms and 5G Technologies," in IEEE Communications Magazine, vol. 56, no. 10, pp. 114-120, OCTOBER 2018, doi: 10.1109/MCOM.2018.1701310.

[42] C. Brewster, I. Roussaki, N. Kalatzis, K. Doolin and K. Ellis, "IoT in Agriculture: Designing a Europe-Wide Large-Scale Pilot," in IEEE Communications Magazine, vol. 55, no. 9, pp. 26-33, Sept. 2017, doi: 10.1109/MCOM.2017.1600528.

Reviewer 3 Report

I think that in its current form, the paper does not provide sufficient material for a possible journal publication.

The authors should add more materials by adding more experiments, explanations, etc. to the manuscript.

Author Response

Thank you for reviewing our paper.

We are of the opinion that the novelty of this work is sufficient for publication. This work brings the ex-vivo measurements for both the antenna performance and the channel characterization i.e., (i) the experimental design and verification of the simulated results on a real horse leg and (ii) characterization of wireless channel and off-horse link budget for network planning for real scenarios. Additionally, (iii) validation measurements are performed within the realistic setting of a barn rather than only considering simulated environments.

We are of the opinion that the paper has already enough material to validate our antenna design. This validation consists of simulation results, phantom measurement in a lab and path loss measurements and measurements of antenna input characteristics (in terms of S11) on a real horse leg in a real horse stable environment. Adding even more experiments would bloat the paper and dilute its message. 

Reviewer 4 Report

The article gives a more or less straightforward solution to a more or less special problem (to design an efficient antenna for a special application). The authors simulate the designed antenna and validate the results with reali life measurements (with a dummy).

The methods and results are presented in a clear way. Although the antenna design would benefit from a second iteration with some fine frequency tuning, given that the final resonance is a little bit off and this kind of (PCB) antenna is easily producable.

Author Response

Thank you for reviewing our paper and your kind words on our report.

The goal of the antenna design was to be robust for the harsh environment. One of the aspects that makes the environment hard is the large near-field variation we can encounter from horse to horse due to the horse hoof morphology, and from environment to environment (grassland, stable concrete floor, hoof lifted which change the underground to air). Thus the goal was not to match it perfectly to one case, but to make sure that we serve the targeted band for as much of possible encountered near-fields that our antenna could encounter while being employed on a horse. This we achieved in our work (see Fig. 5).

The antenna shown in the paper already was a second iteration. First we tuned to the horse’s leg, see the conference papers. The second iteration took into account the housing and the extra element a farrier brings to the antenna nearfield like the filling epoxy. 

Round 2

Reviewer 2 Report

The authors have correctly addressed all my comments and improved the manuscript accordingly.

Reviewer 3 Report

Minor  spell check is required.

Related work can be improved.